# Complete mitochondrial genomes of three vulnerable cave bat species and their phylogenetic relationships within the order Chiroptera

**Michele Molina**[1,2]*, **Guilherme Oliveira**[1,2], **Renato R. M. Oliveira**[1], **Gisele L. Nunes**[1], **Eder S. Pires**[1], **Xavier Prous**[3], **Mariane Ribeiro**[3], **Santelmo Vasconcelos**[1,4]*

1 Instituto Tecnológico Vale, Belém, Pará, Brazil, 2 Programa de Pós-Graduação em Genética e Biologia Molecular, Universidade Federal do Pará, Belém, Pará, Brazil, 3 Gerência de Espeleologia, Vale, Nova Lima, Minas Gerais, Brazil, 4 Programa de Pós-Graduação em Biodiversidade e Evolução, Museu Paraense Emílio Goeldi, Belém, Pará, Brazil

* michele.molina@pq.itv.org (MM); santelmo.vasconcelos@itv.org (SV)

**Data Availability Statement:** The three mitochondrial genomes described were deposited

## Abstract

The IUCN Red List of Threatened Species contains 175 Brazilian bat species that are threatened by extinction in some degree. From this perspective, it is essential to expand the knowledge about the genetic diversity of vulnerable bats. Genomic sequencing can be useful to generate robust and informative genetic references, increasing resolution when analyzing relationships among populations, species, or higher taxonomic levels. In this study, we sequenced and characterized in detail the first complete mitochondrial genomes of *Furipterus horrens*, *Lonchorhina aurita*, and *Natalus macrourus*, and investigated their phylogenetic position based on amino acid sequences of protein-coding genes (PCGs). The mitogenomes of these species are 16,516, 16,697, and 16,668 bp in length, respectively, and each comprises 13 PCGs, 22 tRNA genes, two rRNA genes, and a putative control region (CR). In the three species, genes were arranged similarly to all other previously described bat mitogenomes, and nucleotide composition was also consistent with the reported range. The length and arrangement of rrnS and rrnL were also consistent with those of other bat species, showing a positive AT-skew and a negative GC-skew. Except for trnS1, for which we did not observe the DHU arm, all other tRNAs showed the cloverleaf secondary structure in the three species. In addition, the mitogenomes showed minor differences in start and stop codons, and in all PCGs, codons ending in adenine were more common compared to those ending in guanine. We found that PCGs of the three species use multiple codons to encode each amino acid, following the previously documented pattern. Furthermore, all PCGs are under purifying selection, with atp8 experiencing the most relaxed purifying selection. Considering the phylogenetic reconstruction, *F. horrens* was recovered as sister to *Noctilio leporinus*, *L. aurita* and *Tonatia bidens* shared a node within Phyllostomidae, and *N. macrourus* appeared as sister to Molossidae and Vespertilionidae.

in the GenBank database under the accession numbers MK033190, OR879254 and OR879255.

**Funding:** This work was funded by Vale (Projeto Diversidade Biológica de Cavernas, R100603. CD.0X). Guilherme Oliveira is a CNPq (Conselho Nacional de Desenvolvimento Científico) fellow (307479/2016-1). The funders had no role in study design, data collection and analysis, decision to publish, or preparation of the manuscript.

**Competing interests:** The authors declare no conflict of interest.

## 1. Introduction

The order Chiroptera encompasses a wide range of species with a wide geographic distribution, reflecting their great functional, morphological, and ecological diversities [1]. Due to this diversity, bats play an essential role in the ecosystems they inhabit by providing services such as pollination, seed dispersal, insect control, and organic matter supply to subterranean habitats, contributing to the energy flow of both hypogean and epigean ecosystems [2, 3]. This relationship among bats, karst ecosystems, and their surroundings is important for some species seeking protection against predators and extreme weather conditions in these landscapes, ensuring the maintenance and survival of their populations [4–6].

The Brazilian landscapes are rich in caves, with karst areas accounting for an estimated 7% of the country's surface area [7, 8]. As a result, the country hosts 181 of the 1450 known bat species, or about 13% of the global richness of the group [9–13]. It is important to note that almost half (44%) of bat species in Brazil are classified as trogloxenes, i.e., although they primarily live underground, they also emerge from their habitat searching for food [14]. However, there are anthropogenic interventions that potentially damage cave environments, which are related to important economic activities such as mining, agriculture, and tourism [15–18], posing challenges to bat conservation, as species are directly affected by habitat disturbance.

Thus, since cave systems rely on the allochthonous supply of resources by bats (guano deposition, food scraps, and eventually their carcasses), and bats need these underground habitats for refuge, the functioning of these delicate habitats must be ensured. To achieve this goal, it is necessary to preserve and protect not only the underground physical environment but also the surrounding influence area (or foraging area) and the cave bats themselves. Only by protecting bats can meaningful ecological interactions be maintained in these delicate ecosystems.

However, according to the International Union for Conservation of Nature (IUCN) Red List of Threatened Species, 1,332 Chiroptera species are threatened to some degree [19]. About 13% of these species (175 spp.) occur in Brazil, including cave species such as *Furipterus horrens* [20], *Lonchorhina aurita* [21], and *Natalus macrourus* [22] (identified in the original list as *Natalus espiritosantensis* [23]), which are categorized as vulnerable. *Lonchorhina aurita* remained on the Red List between 2014 and 2022, but it was reclassified as Near Threatened in the current version. These three species are trogloxenes, insectivorous, and occur in the Amazon, being of great importance for the conservation of the karst areas of this important biome.

From this perspective, increasing knowledge of cave bat fauna is essential, especially regarding the endangered species. Thus, genome sequencing approaches based on next-generation technologies can be important tools to support conservation efforts directed to underground ecosystems, such as generating informative genetic references such as complete mitochondrial genomes [24, 25]. Mitogenomes (mtDNA) are relatively small for animal species, generally ranging in length between 13–22 kbp, and can be easily obtained from low-coverage sequencing. This approach is highly cost-effective compared to other strategies for characterizing genetic diversity, while significantly increasing resolution when analyzing relationships among populations, species, or higher taxonomic levels [26].

In this study, we generated and assembled the first complete mitogenomes for the threatened cave bats *F. horrens*, *L. aurita*, and *N. macrourus*, assembling and analyzing in detail the whole sequences of the three species, aiming to increase the genetic data on endangered species in the Brazilian Amazon to support biodiversity conservation strategies in the region. Specifically, we analyzed the nucleotide composition of the entire mtDNAs and examined the codon usage profiles and selective constraints of protein-coding genes (PCGs). We also described the secondary structure of each tRNA and examined the organization of the control region in

**Table 1. Bats sampling locations across the Serra dos Carajás, Pará, Brazil.**

| Species | voucher | Sample location; Coordinates | Mitogenome length (bp) | GenBank accession |
|---|---|---|---|---|
| *Furipterus horrens* | ITV8911 | S11D_0064; -6,392318; -50,313616 | 16,516 | MK033190 |
| *Lonchorhina aurita* | ITV8916 | S11B_0094; -6,3349066; -50,393498 | 16,697 | OR879254 |
| *Natalus macrourus* | ITV8904 | S11B_0094; -6,3349066; -50,393498 | 16,668 | OR879255 |

Voucher numbers, sampling coordinates, mitochondrial genome lengths, and GenBank accession numbers are presented for each analyzed specimen.

detail. Finally, we assessed the phylogenetic position of these three species among other Chiroptera representatives based on the amino acid supermatrix of the 13 PCGs.

## 2. Materials and methods

### 2.1. Sampling and DNA sequencing

Fieldwork was carried out in the Carajás National Forest, in the Brazilian state of Pará, Eastern Amazon (Table 1), in accordance with the sampling permit 49994 granted by ICMBio/MMA (Instituto Chico Mendes de Conservação da Biodiversidade; Brazilian Ministry of Environment). Tissue samples were collected from the dactylopatagium of adult specimens of *F. horrens*, *L. aurita* and *N. macrourus* and preserved in 2 mL cryogenic microtubes containing absolute ethanol, which were kept refrigerated for subsequent laboratory processing. Total genomic DNA was extracted with the DNeasy Blood & Tissue kit (Qiagen), following the manufacturer's instructions for vertebrate tissues. Then, paired-end shotgun libraries were prepared using the Illumina DNA Prep kit (Illumina), according to the manufacturer's instructions, and sequenced on an Illumina NextSeq 500 platform (Illumina, San Diego, CA, USA), using the NextSeq 500/550 v2.5 kit (300 cycles, 2× 150 bp).

### 2.2. Mitochondrial genome assembly, annotation, and analyses

After sequencing, the data were processed using the AdapterRemoval v2 tool [27] with a minimum quality threshold of Phred > 20. Then, the high-quality reads were subjected to a quality check step using the FastQC software (http://bioinformatics.babraham.ac.uk/projects/fastqc/). Circularized mitogenomes were assembled using NOVOPlasty v4.2 [28], with subsequent checking and manual corrections in the software Geneious Prime v2023 (Biomatters). The annotation was performed with the MITOS2 webserver [29], and then gene boundaries were checked and curated in Geneious. The tRNAs and their secondary structure were identified and predicted with MiTFi [30] in MITOS2, being subsequently depicted with the FORNA webserver [31]. Finally, the control region (CR) was also analyzed using the Microsatellite Repeats Finder webserver [32] to detect microsatellite sequences.

### 2.3. Genetic heterogeneity, and genetic diversity analysis

Nucleotide composition was calculated in Geneious, and AT- and GC-skew was computed by using the formulas (A% − T%)/(A% + T%) and (G% − C%)/(G% + C%) [33]. Codon usage and relative synonym codon usage values (RSCU) for each PCG were estimated using the vertebrate mitochondrial code in the Codon Usage webserver (http://www.bioinformatics.org/sms2/codon_usage.html).

To explore selective constraints, the number of synonymous and nonsynonymous substitutions per synonymous and nonsynonymous site (Ks = dS = SS/LS, and Ka = dN = SA/LA) were estimated to calculate the Ka/Ks ($\omega$) ratio in the software MEGA v11 using the Kimura 2-parameter (K2P) model [34], based on pairwise comparisons between each one of the three

sampled species and a representative mitogenome of all the other Chiroptera families with mitogenome data available in GenBank (https://www.ncbi.nlm.nih.gov/genbank/): *Hipposideros larvatus* (Hipposideridae; accession number MN056567), *Rhinolophus affinis* (Rhinolophidae; MT845219), *Mystacina tuberculata* (Mystacinidae; AY960981), *Pteronotus personatus* (Mormoopidae; KU569221), *Myotis albescens* (Vespertilionidae; MF143497), *Tadarida teniotis* (Molossidae; KY581661), *Saccopteryx leptura* (Emballonuridae; KY681816), *Miniopterus fuliginosus* (Miniopteridae; MH523628), *Noctilio leporinus* (Noctilionidae; KU743910), *Lissonycteris angolensis* (Pteropodidae; MN816334), and *Macroderma gigas* (Megadermnatidae; MW006543). The Ka/Ks ratio is used to measure selective pressures acting on a gene, thus, being assumed as neutral when there is a net balance between deleterious and beneficial mutations (Ka/Ks = 1), as positive or under diversifying selection when Ka/Ks > 1, and as negative or under purifying selection when Ka/Ks < 1 [35].

## 2.4. Phylogenetic analyses

In order to reconstruct the phylogenetic relationships within Chiroptera, including the three species analyzed here and all the other bat genera with previously published data, we used the amino acid sequences of all 13 PCGs of a total of 57 bat species (S1 Table). In addition, we included the mitogenomes of species of two other mammalian orders in the analysis as outgroup: *Ailurus fulgens* (Carnivora; MK886830) and *Mogera wogura* (Eulipotyphla; AB099482). Thus, after aligning the amino acid sequences of each gene individually using MAFFT v7.4 [36] with the alignment algorithm set to *Auto* and the JTT200 scoring matrix, the genes were concatenated into a single substitution matrix using Geneious Prime. Then, maximum likelihood (ML) and Bayesian inference (BI) trees were obtained by running RAxML v8.2 [37] and MrBayes v3.2 [38], respectively, as implemented in the CIPRES portal (https://www.phylo.org). The ML analysis was performed using the model PROTGAMMA and 1,000 replicates of rapid bootstrapping, and the BI analysis was performed using four simultaneous runs, each with four Markov chains (T = 0.2) extended through 20,000,000 generations, sampling every 2,000, and using a burn-in fraction of 25% of the trees.

## 3. Results and discussion

Among the analyzed species, *F. horrens* presented the most compact mitogenome, with a total length of 16,516 bp, followed by *L. aurita* and *N. macrourus*, with 16,697 bp and 16,668 bp, respectively (Fig 1). These lengths fall within the range of the complete bat mitogenomes deposited in GenBank up to date, which range from 16,175 bp in *Lasionycteris noctivagans* (MT774149) to 17,660 bp in *Submyotodon moupinensis* (MZ435948). As in the case of all other Chiroptera species, the mitogenomes of *F. horrens*, *L. aurita*, and *N. macrourus* contain the typical set of 37 genes, including 13 PCGs (atp6, atp8, cox1, cox2, cox3, cob, nad1, nad2, nad3, nad4, nad4l, nad5, and nad6), two rRNA genes (rrnS and rrnL), 22 of tRNA (trnA, trnC, trnD, trnE, trnF, trnG, trnH, trnI, trnK, trnL1, trnL2, trnM, trnP, trnQ, trnR, trnS1, trnS2, trnT, trnV, trnW and trnY), and a control region (CR), which varied from 1,092 to 1,247 bp (Table 2).

Vertebrate mitochondrial genomes generally present lower rates of gene rearrangements compared to the nuclear genome, although within some lineages, such as amphibians, crocodilians, fish, lizards, and snakes, species may present higher degrees of structural variation, most of which involving tRNA genes [39–41]. In the three species, the genes were arranged similarly to all other bat species [39, 42, 43], with 29 genes (12 PCGs, the two rRNAs and 14 tRNAs) being located in the heavy (H) strand, while nine genes (nad6 and eight tRNAs) were observed in the light (L) strand (Table 2 and Fig 1). As the overall nucleotide composition

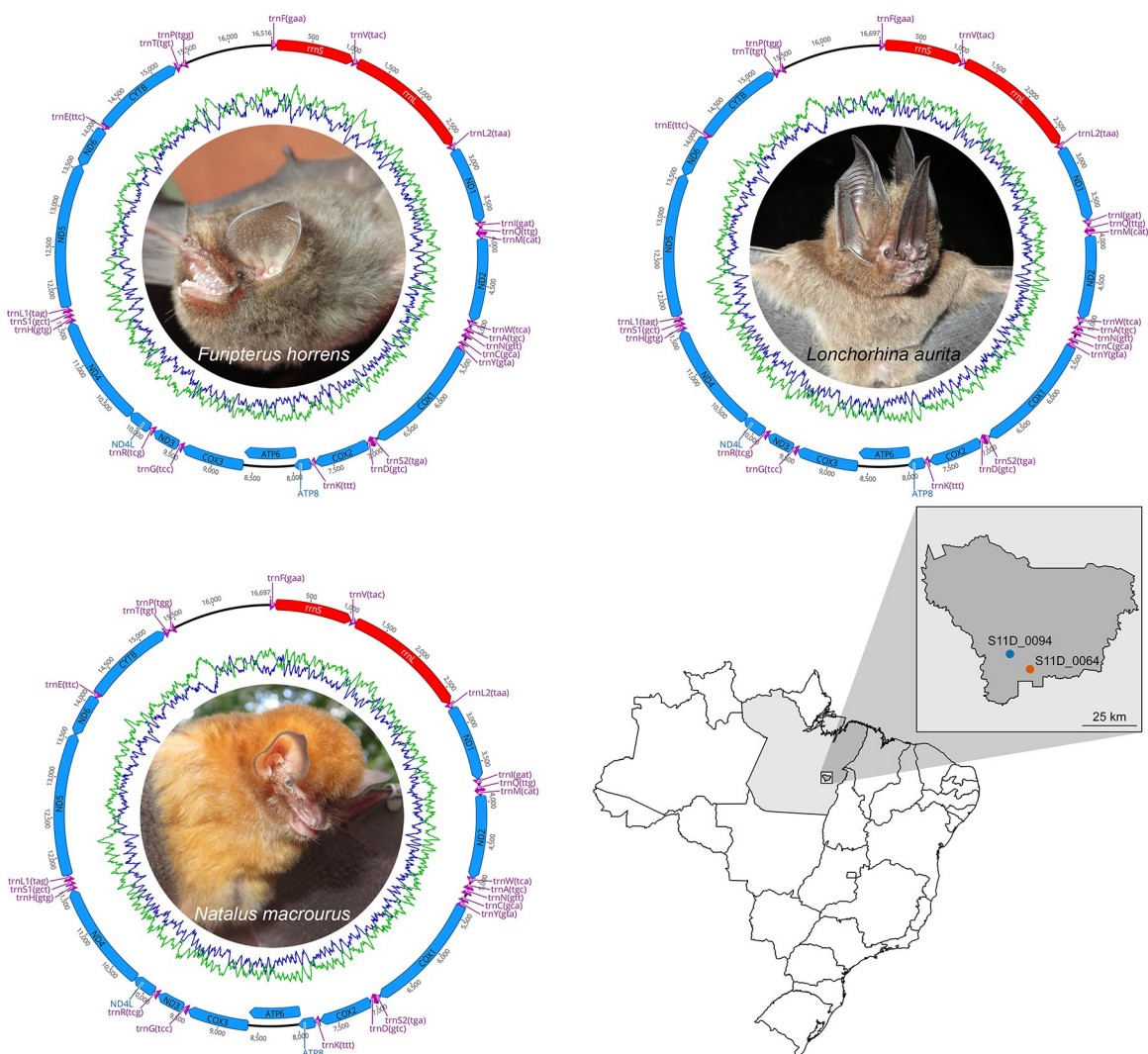

**Fig 1. Circular genome depiction of the mitochondrial genomes of *Furipterus horrens*, *Lonchorhina aurita*, and *Natalus macrourus*, and their sampling locations in the Carajás National Forest, Pará, Brazil.** Colored arrows pointing to the left and right in the circular map of the mitogenomes represent the transcription regions of protein coding genes (blue), rRNA genes (red) and tRNA genes (purple) on the L and H strands, respectively. *Furipterus horrens* was collected in the cave S11D_0064, while both *L. aurita* and *N. macrourus* were collected in the cave S11D_0094.

presented in Table 3, the GC contents of *F. horrens* (44.6%), *L. aurita* (40.5%), and *N. macrourus* (38.6%) were consistent with the range reported for other bats [44, 45], also presenting an AT-skewed nucleotide composition in the PCGs, rRNAs and tRNAs, which is usually favored due to the lower energy cost to break AT bonds during replication and transcription processes in the mitogenome [46]. Furthermore, we observed a strand asymmetry for the three mitogenomes described here, with GC/AT deviations of −0.2421/0.1119 for *F. horrens*, −0.3234/0.0689 for *L. aurita*, and −0.3316/0.1339 for *N. macrourus*, indicating a higher content of pyrimidines in the L strand. The occurrence of asymmetries in mutational patterns may be attributed to replication and transcription mechanisms, which result in strand-specific biases and affect the likelihood of base substitutions between H and L strands, as each strand temporarily exists in a denatured (single-stranded) state, exposing the DNA to damage, thus increasing its susceptibility to mutations [33, 47].

**Table 2. Mitochondrial genomes of *Furipterus horrens* (ITV8911), *Lonchorhina aurita* (ITV8916), and *Natalus macrourus* (ITV8904) indicating their main features.**

| Feature | Strand[a] | *Furipterus horrens* | | | *Lonchorhina aurita* | | | *Natalus macrourus* | | |
|---|---|---|---|---|---|---|---|---|---|---|
| | | Position (Length in bp) | Codons (start/stop) | overlap | Position (Length in bp) | Codons (start/stop) | overlap | Position (Length in bp) | Codons (start/stop) | overlap |
| trnF (GAA) | H | 1–70 (70) | - | 0 | 1–69 (69) | - | 0 | 1–73 (73) | - | 0 |
| rrnS | H | 71–1040 (970) | - | 0 | 70–1044 (975) | - | 0 | 74–1044 (971) | - | 0 |
| trnV (TAC) | H | 1041–1107 (67) | - | 0 | 1045–1113 (69) | - | 0 | 1045–1112 (68) | - | 0 |
| rrnL | H | 1108–2675 (1568) | - | 0 | 1114–2685 (1572) | - | 1 | 1113–2686 (1574) | - | 1 |
| trnL2 (TAA) | H | 2676–2750 (75) | - | 2 | 2687–2761 (75) | - | 2 | 2688–2762 (75) | - | 2 |
| nad1 | H | 2753–3709 (957) | ATG/TAA | -1 | 2764–3720 (957) | ATG/TAA | -1 | 2765–3721 (957) | ATG/TAA | -1 |
| trnI (GAT) | H | 3709–3777 (69) | - | -3 | 3720–3788 (69) | - | -3 | 3721–3789 (69) | - | -3 |
| trnQ (TTG) | L | 3775–3846 (72) | - | 0 | 3786–3858 (73) | - | 0 | 3787–3860 (74) | - | 0 |
| trnM (CAT) | H | 3847–3915 (69) | - | 0 | 3859–3927 (69) | - | 0 | 3861–3929 (69) | - | 0 |
| nad2 | H | 3916–4959 (1044) | ATA/TAG | -2 | 3928–4974 (1047) | ATA/TAG | -2 | 3930–4973 (1044) | ATA/TAG | -2 |
| trnW (TCA) | H | 4958–5025 (68) | - | 3 | 4973–5040 (68) | - | 5 | 4972–5041 (70) | - | 4 |
| trnA (TGC) | L | 5029–5097 (69) | - | 1 | 5046–5114 (69) | - | 0 | 5046–5116 (71) | - | 1 |
| trnN (GTT) | L | 5099–5171 (73) | - | 2 | 5115–5187 (73) | - | 2 | 5193–5223 (31) | - | -1 |
| trnC (GCA) | L | 5205–5270 (66) | - | 0 | 5221–5286 (66) | - | 0 | 5223–5288 (66) | - | 1 |
| trnY (GTA) | L | 5271–5336 (66) | - | 1 | 5287–5353 (67) | - | 1 | 5290–5355 (66) | - | 1 |
| cox1 | H | 5338–6882 (1545) | ATG/TAA | -3 | 5355–6899 (1545) | ATG/TAA | -3 | 5357–6913 (1557) | ATG/TAA | 1 |
| trnS2 (TGA) | L | 6880–6948 (69) | - | 7 | 6897–6965 (69) | - | 7 | 6915–6983 (69) | - | 7 |
| trnD GTC | H | 6956–7022 (67) | - | 0 | 6973–7039 (67) | - | 0 | 6991–7057 (67) | - | 0 |
| cox2 | H | 7023–7710 (688) | ATG/T | -3 | 7040–7723 (684) | ATG/TAA | 3 | 7058–7741 (684) | ATG/TAA | 3 |
| trnK (TTT) | H | 7708–7772 (65) | - | 1 | 7727–7791 (65) | - | 1 | 7745–7811 (67) | - | 1 |
| atp8 | H | 7774–7977 (204) | ATG/TAA | -43 | 7793–7996 (204) | ATG/TAA | -43 | 7813–8016 (204) | ATG/TAA | -43 |
| atp6 | H | 7935–8615 (681) | ATG/TAA | -1 | 7954–8634 (681) | ATG/TAA | -1 | 7974–8654 (681) | ATG/TAA | -1 |
| cox3 | H | 8615–9398 (784) | ATG/T | 0 | 8634–9419 (786) | ATG/TAG | -1 | 8654–9438 (785) | ATG/TA | -1 |
| trnG (TCC) | H | 9399–9467 (69) | - | 0 | 9419–9487 (69) | - | -3 | 9438–9507 (70) | - | -3 |
| nad3 | H | 9468–9815 (348) | ATA/TAA | 0 | 9485–9835 (351) | ATT/TAA | 0 | 9505–9855 (351) | ATT/TAA | 0 |
| trnR (TCG) | H | 9816–9883 (68) | - | 0 | 9836–9903 (68) | - | 0 | 9856–9924 (69) | - | 0 |
| nad4l | H | 9884–10180 (297) | ATG/TAA | -7 | 9904–10200 (297) | ATG/TAA | -7 | 9925–10221 (297) | ATG/TAA | 20 |
| nad4 | H | 10174–1551 (1378) | ATG/T | 0 | 10194–11571 (1378) | ATG/T | 0 | 10242–11622 (1381) | ATG/T | -30 |
| trnH (GTG) | H | 11552–11618 (67) | - | 0 | 11572–11639 (68) | - | 0 | 11593–11659 (67) | - | 0 |
| trnS1 (GCT) | H | 11619–11677 (59) | - | 1 | 11640–11698 (59) | - | 1 | 11660–11718 (59) | - | 1 |

*(Continued)*

**Table 2.** (Continued)

| Feature | Strand[a] | Furipterus horrens | | | Lonchorhina aurita | | | Natalus macrourus | | |
|---|---|---|---|---|---|---|---|---|---|---|
| | | Position (Length in bp) | Codons (start/stop) | overlap | Position (Length in bp) | Codons (start/stop) | overlap | Position (Length in bp) | Codons (start/stop) | overlap |
| trnL1 (TAG) | H | 11679–11749 (71) | - | -9 | 11700–11769 (70) | - | -9 | 11720–11789 (70) | - | -9 |
| nad5 | H | 11741–13567 (1827) | ATA/TAA | -17 | 11761–13587 (1827) | ATA/TAA | -17 | 11781–13610 (1830) | ATA/TAA | -17 |
| nad6 | L | 13551–14075 (525) | ATG/TAA | 3 | 13571–14098 (528) | ATG/TAA | 0 | 13594–14121 (528) | ATG/TAA | 0 |
| trnE (TTC) | L | 14079–14146 (68) | - | 4 | 14099–14171 (69) | - | 4 | 14122–14190 (69) | - | 4 |
| cob | H | 14151–15290 (1140) | ATG/AGA | 0 | 14172–15311 (1140) | ATG/AGA | 0 | 14195–15334 (1140) | ATG/TAA | 0 |
| trnT (TGT) | H | 15291–15357 (67) | - | 0 | 15312–15381 (70) | - | -1 | 15335–15404 (70) | - | -1 |
| trnP (TGG) | L | 15358–15424 (67) | - | - | 15381–15447 (67) | - | - | 15404–15470 (67) | - | - |

[a]H = heavy strand, and L = light strand.

The length and arrangement of the rRNA genes (rrnS and rrnL) in the analyzed mitochondrial genomes also agreed with other bat species, ranging between 970–975 bp, and 1568–1574 bp, respectively. The rrnS gene was observed between trnF and trnV, and the rrnL was located near the rrnS, between trnV and trnL2. In all three species, the two rRNA subunits displayed a positive AT-skew and a negative GC-skew. While the nucleotide composition of rrnS followed an A > C > T > G pattern in the three analyzed mitogenomes, we observed differences in rrnL among the three species. *Furipterus horrens* and *N. macrourus* exhibited an A > T > C > G composition, while *L. aurita* displayed an A > C > T > G pattern.

All tRNA genes in the three species displayed the classic cloverleaf secondary structure, except for trnS1, which differed from other tRNAs by lacking the DHU arm (S1–S3 Figs and S2 Table). The absence of the DHU arm in trnS1 is a conserved trait among metazoans, and the secondary tRNA structures that we modelled agreed with those previously reported for other mammals [30, 42]. The total length of tRNAs in *F. horrens*, *L. aurita*, and *N. macrourus* was between 1502 and 1523 bp, as showed in Table 2. The lengths of individual genes varied from 60 to 75 bp in trnS1 and trnL2, respectively, also agreeing with the lengths reported for other bats [42, 44]. Due to the limited availability of detailed descriptions of tRNA secondary structures in Chiroptera, we provide a comparison among the three species described in this study with *Murina ussuriensis* and *Ectophylla alba*, which were previously analyzed by Yoon & Park [48] and Vivas-Toro et al. [44], respectively. Comparatively, the secondary structures of the tRNAs were very similar among species. However, slight variations in the length of the stem region in certain tRNAs were noted, as reported by Vivas-Toro et al. [44]. Except for trnR, which had a larger variation in length (S3–S5 Tables), the length of the receptor stem was mostly consistent across all tRNAs. On the other hand, the length of the anticodon stem varied among species for three tRNAs: trnL2, trnQ, and trnS1. The TψC stem exhibited a greater variation in length, with variations observed in eight tRNAs: trnF, trnV, trnQ, trnC, trnR, trnH, trnL1, and trnT. Finally, the length of the DHU stem varied in six tRNAs, specifically trnN, trnC, trnS2, trnK, trnT, and trnS1 (trnS1 with the stem being absent as previously mentioned).

The mitogenomes of *F. horrens*, *L. aurita*, and *N. macrourus* showed minor differences in the start and stop codons of the PCGs. In *F. horrens*, only nad2, nad3 and nad5 have ATA as

Table 3. Nucleotide composition of *Furipterus horrens* (ITV8911), *Lonchorhina aurita* (ITV8916), and *Natalus macrourus* (ITV8904).

| Feature | *Furipterus horrens* | | | | | | | | *Lonchorhina aurita* | | | | | | | | *Natalus macrourus* | | | | | | | |
|---|---|---|---|---|---|---|---|---|---|---|---|---|---|---|---|---|---|---|---|---|---|---|---|---|
| | A | C | G | T | AT ratio | GC ratio | AT-skew | GC-skew | A | C | G | T | AT ratio | GC ratio | AT-skew | GC-skew | A | C | G | T | AT ratio | GC ratio | AT-skew | GC-skew |
| PCGs | 31.8 | 28.9 | 12.6 | 26.7 | 58.5 | 41.5 | 0.088 | -0.392 | 30.9 | 28.4 | 11.9 | 28.8 | 59.7 | 40.1 | 0.036 | -0.409 | 34.0 | 27.0 | 11.6 | 27.5 | 61.5 | 38.5 | 0.105 | -0.400 |
| rrnS | 35.4 | 24.4 | 18.1 | 22.1 | 59.6 | 42.5 | 0.232 | -0.148 | 33.7 | 24.0 | 19.1 | 23.2 | 56.9 | 43.1 | 0.186 | -0.114 | 36.9 | 23.1 | 17.1 | 23.0 | 59.9 | 40.2 | 0.232 | -0.149 |
| rrnL | 38.2 | 21.5 | 17.5 | 22.8 | 61.0 | 39.0 | 0.252 | -0.103 | 35.2 | 23.6 | 18.2 | 23.0 | 58.2 | 41.8 | 0.209 | -0.129 | 39.0 | 19.8 | 16.6 | 24.5 | 63.5 | 36.4 | 0.228 | -0.087 |
| CR | 31.7 | 26.9 | 15.8 | 25.6 | 57.3 | 42.7 | 0.105 | -0.213 | 30.8 | 24.9 | 17.4 | 26.9 | 57.7 | 42.3 | 0.068 | -0.144 | 33.1 | 29.4 | 14.8 | 22.7 | 55.8 | 44.2 | 0.188 | -0.282 |

the start codon, whereas the remaining 10 PCGs utilize ATG, as observed in the other two species. In the case of *L. aurita*, we observed ATA as an alternative start codon in nad3 (Table 4). On the other hand, among the PCGs of *F. horrens*, eight PCGs utilize TAA as stop codon, while cox2 and cob use AGA, nad2 uses TAG, and cox3 and nad4 present T as an incomplete stop codon. For *L. aurita*, TAA is used as the stop codon for nine PCGs, while only cob uses AGA, nad2 and cox3 use TAG, and NAD4 presents T as incomplete stop codon (Table 4). Finally, for *N. macrourus*, we found TA and T as incomplete stop codons in cox3 and nad4, respectively, while nad2 presented TAA, and the 10 remaining PCGs with TAA as stop codon, as indicated in Table 4.

Regarding the codon usage bias, it is widely recognized that vertebrates tend to present particular synonymous codons, which can differ depending on the species [49]. In order to assess codon usage bias within the studied species, we used the RSCU method, which involves comparing the observed frequency of a synonymous codon to the expected frequency of that same codon when all codons are used equally for a given amino acid. As previously observed for *E. alba* [44], we noticed that the PCGs of *F. horrens*, *L. aurita*, and *N. macrourus* utilize multiple codons to encode each amino acid, following the pattern already documented for bats, although the codon usage frequency differed among the three studied species, as presented in Table 4. Codons ending in adenine are preferred, while codons ending in guanine are the least used [42, 44]. The most frequently occurring codon among the three species was CTA, which corresponds to the amino acid Leucine (Leu), appearing 186 times in the PCGs of *F. horrens*, 287 of *L. aurita*, and 181 of *N. macrourus* (Table 4). In contrast, the codon CGG for Arginine (Arg) is the least frequently used, being observed seven times in *F. horrens*, once in *L. aurita*, and three times in *N. macrourus* (Table 4).

The ratio of synonymous (Ks) and non-synonymous (Ka) substitutions is a straightforward way to measure the selection pressure on a gene. This ratio indicates neutrality when Ka/Ks = 1, negative selection when Ka/Ks < 1, and positive selection when Ka/Ks > 1, thus aiding to understand evolutionary gene selections [50, 51]. Our analyses indicated that all PCGs of *F. horrens*, *L. aurita* and *N. macrourus* present intense purifying selection, with Ka/Ks < 1. Among the PCGs, atp8 showed the highest Ka/Ks ratio value (0,618 ± 0,123), indicating that it is considerably less affected by selective pressure than the other 12 genes (Fig 2).

Furthermore, the analyzed mitogenomes have overlapping regions between genes, being similar to what has been observed for other mammals (e.g., [52–55]). As previously reported for other bat mitogenomes, we noticed three long overlapping regions in all three species, located between atp8 and atp6, nad5 and nad6, and nad4l and nad4, besides shorter overlaps ranging from 1–3 bp, as detailed in Table 2.

Apart from genes, the organellar genome of animals is also composed of non-coding regions, including the origin of replication (OriL), the D-loop complex (or control region–CR), and a few intergenic spacers, playing a crucial role in the replication and maintenance of the mitochondrial genome [52]. As anticipated, these regions were present in all three species covered in this study. The intergenic spacers of the three species have total lengths of 25, 27, and 29 bp in *F. horrens*, *L. aurita* and *N. macrourus*, respectively, falling within the range of 18–43 bp reported for other bats species in the same taxonomic order, as indicated by Meganathan et al. [42]. According to Seutin et al. [56], OriL is found in the WANCY region (which contains the tRNA genes trnW, trnA, trnN, trnC, and trnY, in that order) between the genes for trnA and trnC in vertebrates. Within Chiroptera, OriL ranges from 31–34 bp, being also identified in three species we analyzed (32 bp in both *F. horrens* and *L. aurita*, and 31 bp in *N. macrourus*).

*Furipterus horrens*, *L. aurita*, and *N. macrourus* have slightly different CR lengths of 1092 bp, 1252 bp, and 1198 bp, respectively. In bats, the CR typically has a GC content slightly

**Table 4. Codon Usage of the 13 PCGs of the mitochondrial genomes of *Furipterus horrens*, *Lonchorhina aurita*, and *Natalus macrourus*.**

| Amino Acid | Codon | *Furipterus horrens* | | | | *Lonchorhina aurita* | | | | *Natalus macrourus* | | | |
|---|---|---|---|---|---|---|---|---|---|---|---|---|---|
| | | Number | /1000 | Fraction | RSCU | Number | /1000 | Fraction | RSCU | Number | /1000 | Fraction | RSCU |
| Ala (A) | GCG | 4 | 1.05 | 0.02 | 0.08 | 4 | 1.05 | 0.02 | 0.08 | 7 | 1.84 | 0.04 | 0.16 |
| | GCA | 56 | 14.7 | 0.27 | 1.08 | 66 | 17.4 | 0.33 | 1.32 | 58 | 15.2 | 0.33 | 1.32 |
| | GCT | 58 | 15.2 | 0.28 | 1.12 | 71 | 18.7 | 0.36 | 1.44 | 44 | 11.6 | 0.25 | 1.00 |
| | GCC | 90 | 23.7 | 0.43 | 1.72 | 58 | 15.3 | 0.29 | 1.16 | 67 | 17.6 | 0.38 | 1.52 |
| Cys (C) | TGT | 17 | 4.47 | 0.53 | 1.06 | 9 | 2.37 | 0.28 | 0.56 | 7 | 1.84 | 0.28 | 0.56 |
| | TGC | 15 | 3.94 | 0.47 | 0.94 | 23 | 6.05 | 0.72 | 1.44 | 18 | 4.73 | 0.72 | 1.44 |
| Asp (D) | GAT | 27 | 7.10 | 0.47 | 0.94 | 46 | 12.1 | 0.53 | 1.06 | 53 | 13.9 | 0.55 | 1.10 |
| | GAC | 30 | 7.88 | 0.53 | 1.06 | 41 | 10.8 | 0.47 | 0.94 | 44 | 11.6 | 0.45 | 0.90 |
| Glu (E) | GAG | 27 | 7.10 | 0.30 | 0.60 | 50 | 13.1 | 0.41 | 0.82 | 39 | 10.2 | 0.35 | 0.70 |
| | GAA | 62 | 16.3 | 0.70 | 1.40 | 71 | 18.7 | 0.59 | 1.18 | 71 | 18.6 | 0.65 | 1.30 |
| Phe (F) | TTT | 57 | 15.0 | 0.38 | 0.76 | 89 | 23.4 | 0.41 | 0.82 | 79 | 20.8 | 0.42 | 0.84 |
| | TTC | 92 | 24.2 | 0.62 | 1.24 | 128 | 33.7 | 0.59 | 1.18 | 110 | 28.9 | 0.58 | 1.16 |
| Gly (G) | GGG | 13 | 3.42 | 0.10 | 0.40 | 17 | 4.47 | 0.11 | 0.44 | 24 | 6.30 | 0.15 | 0.60 |
| | GGA | 48 | 12.6 | 0.35 | 1.40 | 65 | 17.1 | 0.44 | 1.76 | 61 | 6.02 | 0.39 | 1.56 |
| | GGT | 27 | 7.10 | 0.20 | 0.80 | 34 | 8.94 | 0.23 | 0.92 | 38 | 9.98 | 0.24 | 0.96 |
| | GGC | 48 | 12.6 | 0.35 | 1.40 | 32 | 8.41 | 0.22 | 0.88 | 33 | 8.67 | 0.21 | 0.84 |
| His (H) | CAT | 75 | 19.7 | 0.49 | 0.98 | 60 | 15.8 | 0.47 | 0.94 | 83 | 21.8 | 0.49 | 0.98 |
| | CAC | 78 | 20.5 | 0.51 | 1.02 | 68 | 17.9 | 0.53 | 1.06 | 87 | 22.9 | 0.51 | 1.02 |
| Ile (I) | ATT | 110 | 28.9 | 0.49 | 0.98 | 115 | 30.2 | 0.55 | 1.10 | 123 | 32.3 | 0.61 | 1.22 |
| | ATC | 115 | 30.2 | 0.51 | 1.02 | 95 | 25.0 | 0.45 | 0.90 | 80 | 21.0 | 0.39 | 0.78 |
| Lys (K) | AAG | 20 | 5.26 | 0.16 | 0.32 | 20 | 5.26 | 0.19 | 0.38 | 19 | 4.99 | 0.18 | 0.36 |
| | AAA | 102 | 26.8 | 0.84 | 1.68 | 87 | 22.9 | 0.81 | 1.62 | 89 | 23.4 | 0.82 | 1.64 |
| Leu (L) | TTG | 21 | 5.52 | 0.05 | 0.30 | 31 | 8.15 | 0.06 | 0.36 | 28 | 7.35 | 0.06 | 0.36 |
| | TTA | 91 | 23.9 | 0.20 | 1.20 | 116 | 30.5 | 0.23 | 1.38 | 133 | 34.9 | 0.30 | 1.80 |
| | CTG | 42 | 11.0 | 0.09 | 0.54 | 23 | 6.05 | 0.05 | 0.30 | 20 | 5.25 | 0.05 | 0.30 |
| | CTA | 186 | 48.9 | 0.42 | 2.52 | 287 | 49.2 | 0.37 | 2.22 | 181 | 47.5 | 0.41 | 2.46 |
| | CTT | 48 | 12.6 | 0.11 | 0.66 | 64 | 16.8 | 0.13 | 0.78 | 44 | 11.6 | 0.10 | 0.60 |
| | CTC | 57 | 15.0 | 0.13 | 0.78 | 79 | 20.8 | 0.16 | 0.96 | 38 | 9.98 | 0.09 | 0.54 |
| Met (M) | ATG | 51 | 13.4 | 0.26 | 0.52 | 46 | 12.1 | 0.26 | 0.52 | 34 | 8.93 | 0.17 | 0.34 |
| | ATA | 145 | 38.1 | 0.74 | 1.48 | 129 | 33.9 | 0.74 | 1.48 | 163 | 42.8 | 0.83 | 1.66 |
| Asn (N) | AAT | 82 | 21.6 | 0.42 | 0.84 | 61 | 16.0 | 0.48 | 0.96 | 57 | 15.0 | 0.41 | 0.82 |
| | AAC | 111 | 29.2 | 0.58 | 1.16 | 66 | 17.4 | 0.52 | 1.04 | 82 | 21.5 | 0.59 | 1.18 |
| Pro (P) | CCG | 20 | 5.26 | 0.06 | 0.24 | 28 | 7.36 | 0.1 | 0.40 | 17 | 4.46 | 0.07 | 0.28 |
| | CCA | 128 | 33.6 | 0.41 | 1.64 | 96 | 25.2 | 0.33 | 1.32 | 103 | 27.1 | 0.42 | 1.68 |
| | CCT | 88 | 23.1 | 0.28 | 1.12 | 69 | 18.1 | 0.24 | 0.96 | 57 | 15.0 | 0.23 | 0.92 |
| | CCC | 80 | 21.0 | 0.25 | 1.00 | 95 | 25.0 | 0.33 | 1.32 | 70 | 18.4 | 0.28 | 1.12 |
| Gln (Q) | CAG | 55 | 14.5 | 0.39 | 0.78 | 48 | 12.6 | 0.31 | 0.62 | 46 | 12.1 | 0.26 | 0.52 |
| | CAA | 86 | 22.6 | 0.61 | 1.22 | 106 | 27.9 | 0.69 | 1.38 | 133 | 34.9 | 0.74 | 1.48 |
| Arg (R) | CGG | 7 | 1.84 | 0.11 | 0.44 | 1 | 0.26 | 0.02 | 0.08 | 3 | 0.79 | 0.07 | 0.28 |
| | CGA | 24 | 6.31 | 0.38 | 1.52 | 22 | 5.78 | 0.52 | 2.08 | 24 | 6.30 | 0.55 | 2.20 |
| | CGT | 14 | 3.42 | 0.21 | 0.84 | 9 | 2.37 | 0.21 | 0.84 | 5 | 1.31 | 0.11 | 0.44 |
| | CGC | 19 | 4.99 | 0.3 | 1.20 | 10 | 2.63 | 0.24 | 0.96 | 12 | 3.15 | 0.27 | 1.08 |
| Ser (S) | AGT | 17 | 4.47 | 0.05 | 0.30 | 20 | 5.26 | 0.05 | 0.30 | 11 | 2.89 | 0.03 | 0.18 |
| | AGC | 48 | 12.6 | 0.14 | 0.84 | 16 | 4.21 | 0.04 | 0.24 | 28 | 7.35 | 0.08 | 0.48 |
| | TCG | 24 | 6.31 | 0.07 | 0.42 | 27 | 7.10 | 0.07 | 0.42 | 30 | 7.88 | 0.09 | 0.54 |
| | TCA | 98 | 25.8 | 0.28 | 1.68 | 114 | 30.0 | 0.3 | 1.80 | 116 | 30.5 | 0.34 | 2.04 |

*(Continued)*

**Table 4.** (Continued)

| Amino Acid | Codon | *Furipterus horrens* | | | | *Lonchorhina aurita* | | | | *Natalus macrourus* | | | |
|---|---|---|---|---|---|---|---|---|---|---|---|---|---|
| | | Number | /1000 | Fraction | RSCU | Number | /1000 | Fraction | RSCU | Number | /1000 | Fraction | RSCU |
| | TCT | 87 | 22.9 | 0.25 | 1.50 | 88 | 23.1 | 0.23 | 1.38 | 65 | 17.1 | 0.19 | 1.14 |
| | TCC | 80 | 21.0 | 0.23 | 1.38 | 119 | 31.3 | 0.31 | 1.86 | 92 | 24.2 | 0.27 | 1.62 |
| Thr (T) | ACG | 22 | 5.78 | 0.06 | 0.24 | 25 | 6.57 | 0.09 | 0.36 | 25 | 6.57 | 0.08 | 0.32 |
| | ACA | 128 | 33.6 | 0.34 | 1.36 | 101 | 26.6 | 0.35 | 1.40 | 120 | 31.5 | 0.39 | 1.56 |
| | ACT | 101 | 26.5 | 0.27 | 1.08 | 66 | 17.4 | 0.23 | 0.92 | 54 | 14.2 | 0.18 | 0.72 |
| | ACC | 123 | 32.3 | 0.33 | 1.32 | 94 | 24.7 | 0.33 | 1.32 | 106 | 27.8 | 0.35 | 1.40 |
| Val (V) | GTG | 15 | 3.94 | 0.11 | 0.44 | 17 | 4.47 | 0.12 | 0.48 | 18 | 4.73 | 0.12 | 0.48 |
| | GTA | 63 | 16.6 | 0.47 | 1.88 | 60 | 15.8 | 0.42 | 1.68 | 60 | 15.8 | 0.4 | 1.60 |
| | GTT | 32 | 8.41 | 0.24 | 0.96 | 35 | 9.20 | 0.24 | 0.96 | 49 | 12.9 | 0.33 | 1.32 |
| | GTC | 25 | 6.57 | 0.19 | 0.76 | 31 | 8.15 | 0.22 | 0.88 | 23 | 6.04 | 0.15 | 0.60 |
| Trp (W) | TGG | 16 | 4.20 | 0.20 | 0.40 | 11 | 2.89 | 0.17 | 0.34 | 12 | 3.15 | 0.18 | 0.36 |
| | TGA | 65 | 17.1 | 0.80 | 1.60 | 53 | 13.9 | 0.83 | 1.66 | 55 | 14.4 | 0.82 | 1.64 |
| Tyr (Y) | TAT | 89 | 23.4 | 0.48 | 0.96 | 109 | 28.7 | 0.54 | 1.08 | 125 | 32.8 | 0.52 | 1.04 |
| | TAC | 96 | 25.2 | 0.52 | 1.04 | 93 | 24.5 | 0.46 | 0.92 | 115 | 30.2 | 0.48 | 0.96 |
| Stop (*) | AGG | 18 | 4.73 | 0.12 | 0.48 | 2 | 0.53 | 0.01 | 0.04 | 2 | 0.53 | 0.01 | 0.04 |
| | AGA | 14 | 3.68 | 0.09 | 0.36 | 8 | 2.10 | 0.04 | 0.16 | 2 | 0.53 | 0.01 | 0.04 |
| | TAG | 51 | 13.4 | 0.34 | 1.36 | 79 | 20.8 | 0.42 | 1.68 | 94 | 24.7 | 0.43 | 1.72 |
| | TAA | 68 | 17.9 | 0.45 | 1.80 | 101 | 26.6 | 0.53 | 2.12 | 122 | 32.0 | 0.55 | 2.20 |

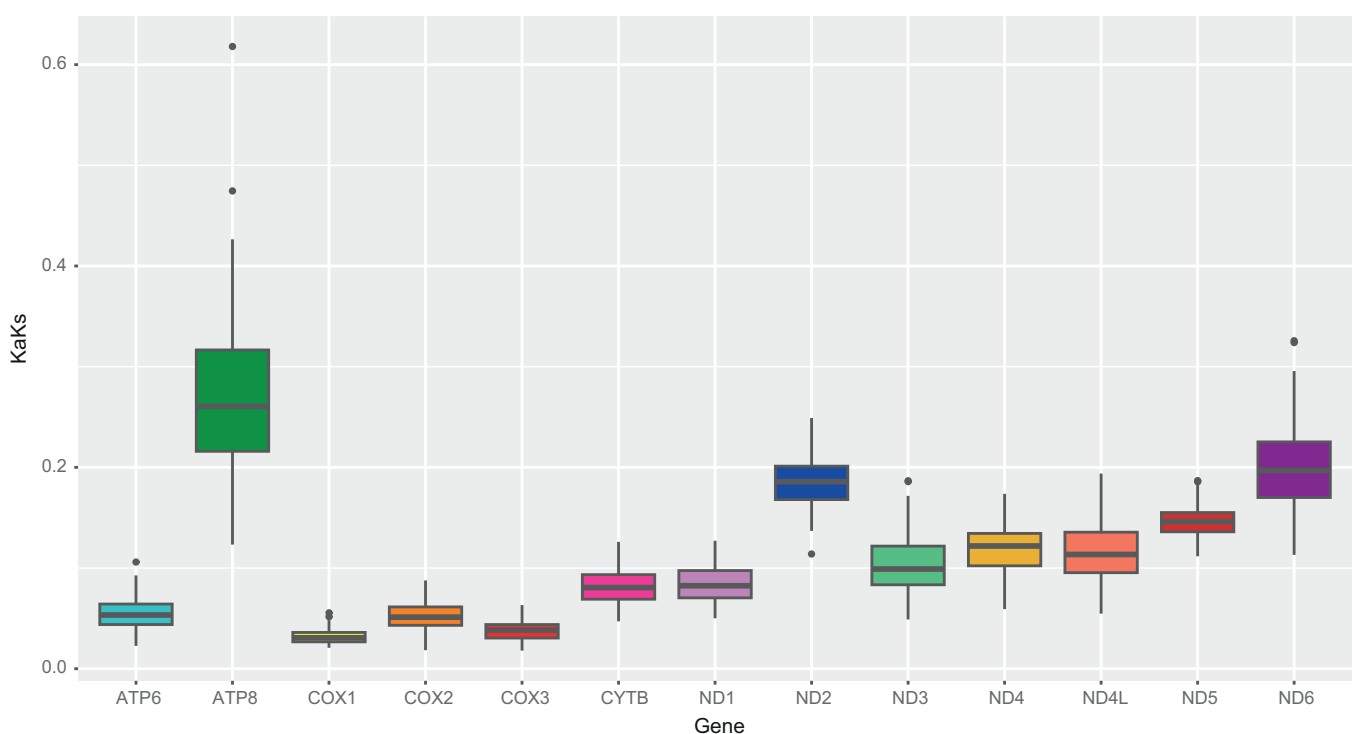

**Fig 2. Selective pressure analysis in the protein coding genes of *Furipterus horrens*, *Lonchorhina aurita*, and *Natalus macrourus*.** The Ka/Ks ratio variation is presented for each protein-coding gene.

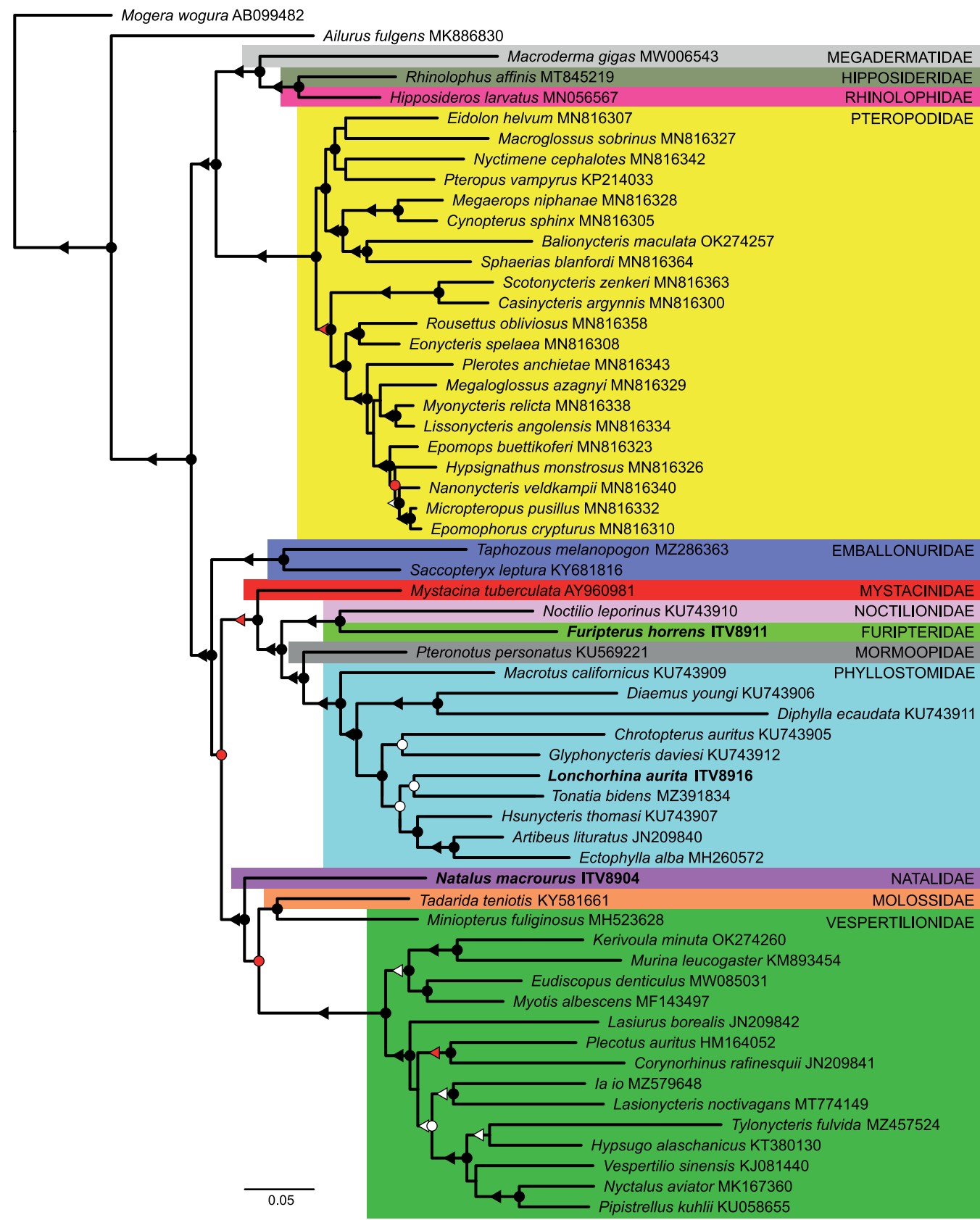

**Fig 3. Phylogenetic relationships among Chiroptera genera using mitogenome protein-coding genes.** Majority-rule consensus tree based on Bayesian inference evidencing the phylogenetic relationships among the mitogenomes of *Furipterus horrens*, *Lonchorhina aurita*, *Natalus macrourus*, and 54 other bat genera, with one species each, indicating their respective family affiliations and GenBank accession numbers. The circles on the nodes and triangles over the branches indicate the following posterior probabilities (PP) and bootstrap (BS) supports, respectively: black means PP = 1.00 and BS ≥ 90%; red is for PP between 0.95–0.99 and BS between 80–89%; and white is for PP between 0.90–0.94 and BS between 70–79%.

below the AT content, as also observed in all three mitogenomes (*F. horrens* with 42.7%, *L. aurita* with 42.3%, and *N. macrourus* with 44.2%). Considering the repetitive portions of the mitogenomes, we observed eight, 13, and 10 microsatellites for *F. horrens*, *L. aurita*, and *N. macrourus*, respectively. Most of these microsatellites have repeated sequences of TA or AT dinucleotides (S3–S5 Tables). Furthermore, the three functional domains (extended terminal associated sequences [ETAS], core, and conserved sequence block [CSB]) present in other bats and most mammals [44, 57, 58] were also observed in the CR of *F. horrens*, *L. aurita*, and *N. macrourus*.

Regarding the phylogenetic reconstructions, the Chiroptera species were grouped into two major clades, with the first including Megadermatidae, Hipposideridae, Rhinolophidae and Pteropodidae, and the second being composed of Emballonuridae, Mystacinidae, Noctilionidae, Furipteridae, Mormoopidae, Phyllostomidae, Natalidae, Molossidae and Vespertilionidae (Fig 3). In general, the relationships among and within families were well resolved, with most of the groups presenting strong support values (Fig 3). The most notable exception occurred between Molossidae and Vespertilionidae, which were recovered as paraphyletic, although with relevant support only in the Bayesian inference, with *Tadarida teniotis* (Molossidae) and *Miniopterus fulilginosus* (Vespertilionidae) appearing as sister lineages (Fig 3). A conflicting pattern involving both families was previously observed in the topology based on transcriptome data presented by Lei & Dong [59]. Considering the phylogenetic position of the three new mitogenomes, *F. horrens* (Furipteridae) was recovered as sister to *Noctilio leporinus* (Noctilionidae) with a strong support, *L. aurita* and *Tonatia bidens* shared a node with moderate support within Phyllostomidae, and *N. macrourus* (Natalidae) appeared as sister to the group formed by Molossidae and Vespertilionidae species (Fig 3).

## 4. Conclusion

In this study, we sequenced and analyzed the complete mitochondrial genomes of three endangered cave bat species, *F. horrens*, *L. aurita*, and *N. marourus*. We studied in detail these organellar genomes, presenting their nucleotide composition, bias and codon usage, the secondary structure of tRNAs, and CR characteristics of these species. The assembly and detailed analysis of the complete mitochondrial genomes may be helpful for phylogenetic studies, helping to identify species in monitoring approaches such as those based on eDNA sampling. The characterization of genetic resources represents a step forward in efforts to conserve bat species, especially endangered ones, as they allow measuring and managing the genetic diversity of populations. Thus, the comprehensive mitogenome analysis and the several genetic markers we described for the three mitogenomes will be useful for future studies on the evolutionary patterns of the populations of the three species.

## Supporting information

**S1 Fig. Secondary structure of the 22 tRNAs in the mitochondrial genome of *Furipterus horrens*.**
(PDF)

**S2 Fig. Secondary structure of the 22 tRNAs in the mitochondrial genome of *Lonchorhina aurita*.**
(PDF)

**S3 Fig. Secondary structure of the 22 tRNAs in the mitochondrial genome of *Natalus macrourus*.**
(PDF)

**S1 Table. Chiroptera species used in the phylogenetic reconstructions with whole mitogenome sequences indicating their respective family affiliations and GenBank accession numbers.**
(DOCX)

**S2 Table. Comparison of the length of the four stems that conform the secondary structure of the tRNAs of *Furipterus horrens*, *Lonchorhina aurita*, and *Natalus macrourus*.**
(DOCX)

**S3 Table. Microsatellites sequences found in the CR of the mitochondrial genome of *Furipterus horrens*.**
(DOCX)

**S4 Table. Microsatellites sequences found in the CR of the mitochondrial genome of *Lonchorhina aurita*.**
(DOCX)

**S5 Table. Microsatellites sequences found in the CR of the mitochondrial genome of *Natalus macrourus*.**
(DOCX)

## Acknowledgments

The authors would like to thank the laboratory technician Manoel Lopes.

## Author Contributions

**Conceptualization:** Guilherme Oliveira, Xavier Prous, Mariane Ribeiro, Santelmo Vasconcelos.

**Data curation:** Michele Molina, Renato R. M. Oliveira, Gisele L. Nunes, Santelmo Vasconcelos.

**Formal analysis:** Michele Molina, Renato R. M. Oliveira, Gisele L. Nunes, Santelmo Vasconcelos.

**Funding acquisition:** Guilherme Oliveira, Santelmo Vasconcelos.

**Investigation:** Renato R. M. Oliveira, Gisele L. Nunes, Eder S. Pires, Xavier Prous, Santelmo Vasconcelos.

**Methodology:** Michele Molina, Guilherme Oliveira, Gisele L. Nunes, Eder S. Pires, Xavier Prous, Mariane Ribeiro, Santelmo Vasconcelos.

**Project administration:** Santelmo Vasconcelos.

**Resources:** Guilherme Oliveira.

**Supervision:** Guilherme Oliveira.

**Validation:** Michele Molina, Renato R. M. Oliveira, Santelmo Vasconcelos.

**Visualization:** Michele Molina, Renato R. M. Oliveira, Gisele L. Nunes, Santelmo Vasconcelos.

**Writing – original draft:** Michele Molina, Santelmo Vasconcelos.

**Writing – review & editing:** Michele Molina, Guilherme Oliveira, Renato R. M. Oliveira, Gisele L. Nunes, Eder S. Pires, Xavier Prous, Mariane Ribeiro, Santelmo Vasconcelos.

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
