## [Decision Letter · Decision Letter 0]

27 Mar 2024

PONE-D-24-05101Complete mitochondrial genome of three vulnerable cave bat speciesPLOS ONE

Dear Dr. Vasconcelos,

Thank you for submitting your manuscript to PLOS ONE. After careful consideration, we feel that it has merit but does not fully meet PLOS ONE’s publication criteria as it currently stands. Therefore, we invite you to submit a revised version of the manuscript that addresses the points raised during the review process. Please submit your revised manuscript by May 11 2024 11:59PM. If you will need more time than this to complete your revisions, please reply to this message or contact the journal office at plosone@plos.org. Please include the following items when submitting your revised manuscript:A rebuttal letter that responds to each point raised by the academic editor and reviewer(s). You should upload this letter as a separate file labeled 'Response to Reviewers'.A marked-up copy of your manuscript that highlights changes made to the original version. You should upload this as a separate file labeled 'Revised Manuscript with Track Changes'.An unmarked version of your revised paper without tracked changes. You should upload this as a separate file labeled 'Manuscript'.

We look forward to receiving your revised manuscript.

Kind regards,

Pankaj Bhardwaj, Ph.D.

Academic Editor

PLOS ONE

Journal Requirements:

   "This work was funded by Vale (Projeto Diversidade Biológica de Cavernas, R100603.CD.0X). Guilherme Oliveira is a CNPq (Conselho Nacional de Desenvolvimento Científico) fellow (307479/2016-1)."

Reviewers' comments:

Reviewer's Responses to Questions

**Comments to the Author**

1. Is the manuscript technically sound, and do the data support the conclusions?

Reviewer #1: Yes

Reviewer #2: Yes

Reviewer #3: Yes

2. Has the statistical analysis been performed appropriately and rigorously? 

Reviewer #1: Yes

Reviewer #2: Yes

Reviewer #3: Yes

3. Have the authors made all data underlying the findings in their manuscript fully available?

Reviewer #1: Yes

Reviewer #2: Yes

Reviewer #3: Yes

4. Is the manuscript presented in an intelligible fashion and written in standard English?

Reviewer #1: Yes

Reviewer #2: Yes

Reviewer #3: No

5. Review Comments to the Author

Reviewer #1: Dear Editor,

The manuscript by Molina et al. presents three new complete mitogenome sequences along with a comprehensive datasets encompassing 57 bat species, facilitating the reconstruction of phylogenies based on Bayesian inference (BI) and maximum likelihood (ML) analyses using the amino acid sequences of 13 protein-coding genes (PCG).

While the manuscript introduces a novel idea and represents a fresh contribution to this field, I have only a few minor suggestions that should be address in the revise version:

The title should be improved. This is not only about new sequence but also the thorough analyses of all known sequences available on this order. So, the phylogeny implication should add in the title.

"Complete mitochondrial genome of three vulnerable cave bat species with phylogenetic implications for the order Chiroptera"

1.Kindly discuss about the genome arrangement in the complete mitochondrial genome. As the mitogenomes of some animal groups exhibit novel rearrangements in gene. Is there any such rare rearrangements exist so far in this order?

2.Page 15 [Line 10] we observed eight, 13, and 10 microsatellites

13? Please compare with Table S4.

3.Please provide a table with species name, accession number, genome size and reference.

Regards

Reviewer #2: The authors sequenced and characterized the complete mitochondrial genomes of three cave bat species, and constructed a phylogenetic tree. The results show that the structure, length, and base composition of the three mitochondrial genomes are conserved. The manuscript as a whole is commendable. My main concern is that the figures and tables presented in this manuscript are difficult to read and understand.

Minor concerns

-Table 1: It is recommended to use a sampling map instead of a table.

-Table 2: The overlap between trnP and trnF is 656 bp? Please check and verify all data.

-Table 3: The table is incomplete.

-Fig. 1: What does “la io” mean.

-Fig. 2: The resolution of Figure 2 can be improved.

-Line 1: Should be “genomes…”, not “genome…”.

-Line 133: Is it GC skew or GC-skew?

-Line 159: Is the phylogenetic tree constructed using 13 amino acid sequences or 13 PCGs? According to line 177, the tree was constructed using the PCGs.

-Lines 178-179: Replace “.” with “,”.

Reviewer #3: Comments to the Author

In this manuscript, the authors describe the sequencing of the complete mitochondrial genome of three vulnerable cave bat species (Furipterus horrens, Lonchorhina aurita, and Natalus macrourus). They describe the mitochondrial gene order of these species, which is the same as that of all other bats. The authors provide detailed descriptions of the tRNAs and provide a particularly nice description of the structure of the mitochondrial control region. They also discuss the predicted amino acid composition of the mitochondrial protein coding genes for the three species. Finally, the authors conduct a phylogenetic analysis of mitogenomes from Chiroptera genera, including 54 previously published mitogenomes of bat species, two previously published mitogenomes from outgroup species, and three newly determined mitogenomes in this study. Some comments as below:

1. The results of the phylogenetic analysis should be shown in "Abstract".

2. The table header is missed in "Table 3".

3. In the list of references, the format of the journal names is inconsistent. Please check and make revisions as required by the journal format.

4. The English language of this manuscript needs to be revised for clarity. For example:

(1) Line 28, Suggested reword "and each comprises......"

(2) Line 96, Suggested reword "......Finally, we assessed the"

(3) Line 98, Suggested reword "on the amino acid supermatrix of the 13 PCGs."

(4) Line 127, Suggested reword "checked and curated in Geneious. The tRNAs ......"

(5) Line 171, Suggested reword "...... the BI tree was"

(6) Line 178, Suggested reword "...... of Furipterus horrens, "

(7) Line 179, Suggested reword "Lonchorhina aurita, Natalus macrourus, and 54 other bat genera with one species each, "

(8) Line 181, Suggested reword "......black PP = 1.00; red "

(9) Line 182, Suggested reword "......and white 0.95 > PP ≥ 0.90. The triangles ...... "

(10) Line 183, Suggested reword "...... approach: black"

(11) Line 324, Suggested reword "The Chiroptera was grouped ......"

6. PLOS authors have the option to publish the peer review history of their article (what does this mean?). If published, this will include your full peer review and any attached files.

Reviewer #1: **Yes: **Dr. Muhammad Asghar Hassan (Institute of Entomology, Guizhou University, Guiyang, China)

Reviewer #2: No

Reviewer #3: No

---

## [Author Response · Author response to Decision Letter 0]

11 May 2024

To 

Dr. Pankaj Bhardwaj 

PLoS One

Academic Editor

Dear Editor,

We are grateful for the reviews we received, which undoubtedly will contribute to enhancing our work. We have revised the manuscript thoroughly and the text was modified according to the comments of the reviewers. All featured comments are listed and answered below.

REVIEWER #1 

The title should be improved. This is not only about new sequence but also the thorough analyses of all known sequences available on this order. So, the phylogeny implication should add in the title.

"Complete mitochondrial genome of three vulnerable cave bat species with phylogenetic implications for the order Chiroptera"

We have changed the title to “Complete mitochondrial genomes of three vulnerable cave bat species and their phylogenetic relationships within the order Chiroptera”.

1. Kindly discuss about the genome arrangement in the complete mitochondrial genome. As the mitogenomes of some animal groups exhibit novel rearrangements in gene. Is there any such rare rearrangements exist so far in this order?

We have included the suggested discussion in the beginning of the second paragraph of the “Results and discussion” section.

2. Page 15 [Line 10] we observed eight, 13, and 10 microsatellites

13? Please compare with Table S4.

We have corrected this information in the text and the respective Supplementary Tables.

3. Please provide a table with species name, accession number, genome size and reference.

We have included the GenBank accession numbers and mitogenome sizes in the Table 1.

REVIEWER #2

-Table 1: It is recommended to use a sampling map instead of a table.

We have included a map with the sampling points in the Figure 1 (former Figure 2), alongside the mitogenome depictions of the three species.

-Table 2: The overlap between trnP and trnF is 656 bp? Please check and verify all data.

We have checked the data and corrected this information in Table 2.

-Table 3: The table is incomplete.

We have corrected Table 3 putting the headings that were missing.

-Fig. 1: What does “la io” mean.

Ia io (great evening bat) is a species from southeastern Asia that is included in the family Vespertilionidae.

-Fig. 2: The resolution of Figure 2 can be improved.

As it contains photos of the bats, Figure 2 is a bitmap image. Due to file size and format restrictions, its resolution was severely lowered in the upload process to the submission platform. We agree that the resolution as reviewed was inappropriate, and we will provide the figure with a proper resolution once we can upload the file.

-Line 1: Should be “genomes…”, not “genome…”.

We have modified the text accordingly.

-Line 133: Is it GC skew or GC-skew?

We have modified the text accordingly, using GC-skew in all mentions.

-Line 159: Is the phylogenetic tree constructed using 13 amino acid sequences or 13 PCGs? According to line 177, the tree was constructed using the PCGs. 

The phylogenetic analyses were performed with the amino acid sequences of the 13 PCGs. We have modified the text to clarify the points mentioning the matrix used in the phylogenetic reconstructions.

-Lines 178-179: Replace “.” with “,”.

We have modified the text accordingly.

REVIEWER #3

1. The results of the phylogenetic analysis should be shown in "Abstract".

As requested, we have included the part of the phylogenetic analysis in the abstract.

2. The table header is missed in "Table 3".

We have corrected Table 3 putting the headings that were missing.

3. In the list of references, the format of the journal names is inconsistent. Please check and make revisions as required by the journal format.

We have checked and corrected the references, according to the journal format.

4. The English language of this manuscript needs to be revised for clarity. For example:

(1) Line 28, Suggested reword "and each comprises......"

(2) Line 96, Suggested reword "......Finally, we assessed the"

(3) Line 98, Suggested reword "on the amino acid supermatrix of the 13 PCGs."

(4) Line 127, Suggested reword "checked and curated in Geneious. The tRNAs ......"

(5) Line 171, Suggested reword "...... the BI tree was"

(6) Line 178, Suggested reword "...... of Furipterus horrens, "

(7) Line 179, Suggested reword "Lonchorhina aurita, Natalus macrourus, and 54 other bat genera with one species each, "

(8) Line 181, Suggested reword "......black PP = 1.00; red "

(9) Line 182, Suggested reword "......and white 0.95 > PP ≥ 0.90. The triangles ...... "

(10) Line 183, Suggested reword "...... approach: black"

(11) Line 324, Suggested reword "The Chiroptera was grouped ......"

We have carefully revised the manuscript and corrected all issues related to the English language, including all points mentioned by the reviewer. 

In addition to the corrections indicated by the reviewers, we also included other modifications, which are also highlighted in the text, aiming to improve the manuscript.

Thank you very much for your kind attention.

Sincerely yours,

Santelmo Vasconcelos

Instituto Tecnológico Vale

Rua Boaventura da Silva 955

66055-090 Belém, Pará, Brazil

Phone: +55 91 3213 5400

santelmo.vasconcelos@itv.org

---

## [Decision Letter · Decision Letter 1]

7 Jun 2024

PONE-D-24-05101R1Complete mitochondrial genomes of three vulnerable cave bat species and their phylogenetic relationships within the order ChiropteraPLOS ONE

Dear Dr. Vasconcelos,

Thank you for submitting your manuscript to PLOS ONE. After careful consideration, we feel that it has merit but does not fully meet PLOS ONE’s publication criteria as it currently stands. Therefore, we invite you to submit a revised version of the manuscript that addresses the points raised during the review process.

We look forward to receiving your revised manuscript.

Kind regards,

Pankaj Bhardwaj, Ph.D.

Academic Editor

PLOS ONE

Journal Requirements:

Reviewers' comments:

Reviewer's Responses to Questions

**Comments to the Author**

1. If the authors have adequately addressed your comments raised in a previous round of review and you feel that this manuscript is now acceptable for publication, you may indicate that here to bypass the “Comments to the Author” section, enter your conflict of interest statement in the “Confidential to Editor” section, and submit your "Accept" recommendation.

Reviewer #1: All comments have been addressed

Reviewer #3: All comments have been addressed

2. Is the manuscript technically sound, and do the data support the conclusions?

Reviewer #1: Yes

Reviewer #3: Yes

3. Has the statistical analysis been performed appropriately and rigorously? 

Reviewer #1: Yes

Reviewer #3: Yes

4. Have the authors made all data underlying the findings in their manuscript fully available?

Reviewer #1: Yes

Reviewer #3: Yes

5. Is the manuscript presented in an intelligible fashion and written in standard English?

Reviewer #1: Yes

Reviewer #3: Yes

6. Review Comments to the Author

**Reviewer #1: **Dear Editor,

I agree with the current version of the manuscript, which is much improved, and all comments have been thoroughly addressed.

Regards,

**Reviewer #3: **1. Line 76, delete "-" between "Natalus espiritosantensis" and "[23]"

2. Line 77, replace "Lonchorhina" with "L."

3. Line 105, The results of three new mitogenomes should been shown in "3. Results and discussion".

4. Line 152, Please add the full title of K2P.

5. Line 168, delete "mtDNA"

6. Line 202-203, Suggested replace "PCGs (ATP6, ATP8, COX1, COX2, COX3, CYTB, ND1, ND2, ND3, ND4, ND4L, ND5, and ND6), two rRNA genes [12S (rrnS) and 16S rRNA (rrnL)]" wuth "PCGs (atp6, atp8, cox1, cox2, cox3, cob, nad1, nad2, nad3, nad4, nad4l, nad5, and nad6), two rRNA genes (rrnS) and rrnL)". At the same time, the corresponding gene names should be corrected elsewhere in this manuscript.

7. Line 312, What does "WANCY" mean?

8. Line 316, replace "Furipterus" with "F."

9. Line 326, replace "mitogenomes" with "species"

10. Line 413, replace "Cuvier, F" with "Cuvier F"

11. Line 415, replace "Tomes, R. F" with "Tomes RF"

12. Line 417, replace "Gervais, P" with "Gervais P"

7. PLOS authors have the option to publish the peer review history of their article (what does this mean?). If published, this will include your full peer review and any attached files.

Reviewer #1: **Yes: **Muhammad Asghar Hassan (Institute of Entomology, Guizhou University, Guiyang, China)

Reviewer #3: No

---

## [Author Response · Author response to Decision Letter 1]

21 Jun 2024

To 

Dr. Pankaj Bhardwaj 

PLoS One

Academic Editor

Dear Editor,

We are grateful for the reviews we received, which undoubtedly will contribute to enhancing our work. We have revised the manuscript thoroughly and the text was modified according to the comments of the reviewers. All featured comments are listed and answered below, including those from the first revision round.

2nd Revision round

REVIEWER #3

1. Line 76, delete "-" between "Natalus espiritosantensis" and "[23]"

- We have modified the text accordingly.

2. Line 77, replace "Lonchorhina" with "L."

- Since abbreviating the genus name at the beginning of the sentence is not recommended, we preferred to keep it in full, as in the case of the 8th comment of Reviewer #3 for Furipterus (at the line 316).

3. Line 105, The results of three new mitogenomes should been shown in "3. Results and discussion".

- We have modified the text accordingly, also changing the citation of the phylogenetic tree to the “Results and discussion” section and including a supplementary table listing the mitogenomes used in the phylogenetic reconstruction in the “Materials and methods” section (2.4 Phylogenetic analyses).

4. Line 152, Please add the full title of K2P.

- We have modified the text accordingly.

5. Line 168, delete "mtDNA"

- We have modified the text accordingly.

6. Line 202-203, Suggested replace "PCGs (ATP6, ATP8, COX1, COX2, COX3, CYTB, ND1, ND2, ND3, ND4, ND4L, ND5, and ND6), two rRNA genes [12S (rrnS) and 16S rRNA (rrnL)]" wuth "PCGs (atp6, atp8, cox1, cox2, cox3, cob, nad1, nad2, nad3, nad4, nad4l, nad5, and nad6), two rRNA genes (rrnS) and rrnL)". At the same time, the corresponding gene names should be corrected elsewhere in this manuscript.

- We have modified the text accordingly.

7. Line 312, What does "WANCY" mean?

- WANCY is the region comprising the tRNA genes trnW, trnA, trnN, trnC and trnY, located between the protein coding genes nad2 and cox1, commonly observed in vertebrate mitogenomes. We have modified the text to clarify the meaning of the word.

8. Line 316, replace "Furipterus" with "F."

- As explained in the response to the 2nd comment, the genus name is in the beginning of the paragraph, and we preferred to keep it in full.

9. Line 326, replace "mitogenomes" with "species"

- We have modified the text accordingly.

10. Line 413, replace "Cuvier, F" with "Cuvier F"

- We have modified the text accordingly.

11. Line 415, replace "Tomes, R. F" with "Tomes RF"

- We have modified the text accordingly.

12. Line 417, replace "Gervais, P" with "Gervais P"

- We have modified the text accordingly.

1st Revision round

REVIEWER #1 

The title should be improved. This is not only about new sequence but also the thorough analyses of all known sequences available on this order. So, the phylogeny implication should add in the title.

"Complete mitochondrial genome of three vulnerable cave bat species with phylogenetic implications for the order Chiroptera"

- We have changed the title to “Complete mitochondrial genomes of three vulnerable cave bat species and their phylogenetic relationships within the order Chiroptera”.

1. Kindly discuss about the genome arrangement in the complete mitochondrial genome. As the mitogenomes of some animal groups exhibit novel rearrangements in gene. Is there any such rare rearrangements exist so far in this order?

- We have included the suggested discussion in the beginning of the second paragraph of the “Results and discussion” section.

2. Page 15 [Line 10] we observed eight, 13, and 10 microsatellites

13? Please compare with Table S4.

- We have corrected this information in the text and the respective Supplementary Tables.

3. Please provide a table with species name, accession number, genome size and reference.

- We have included the GenBank accession numbers and mitogenome sizes in the Table 1.

REVIEWER #2

-Table 1: It is recommended to use a sampling map instead of a table.

- We have included a map with the sampling points in the Figure 1 (former Figure 2), alongside the mitogenome depictions of the three species.

-Table 2: The overlap between trnP and trnF is 656 bp? Please check and verify all data.

- We have checked the data and corrected this information in Table 2.

-Table 3: The table is incomplete.

- We have corrected Table 3 putting the headings that were missing.

-Fig. 1: What does “la io” mean.

- Ia io (great evening bat) is a species from southeastern Asia that is included in the family Vespertilionidae.

-Fig. 2: The resolution of Figure 2 can be improved.

- As it contains photos of the bats, Figure 2 is a bitmap image. Due to file size and format restrictions, its resolution was severely lowered in the upload process to the submission platform. We agree that the resolution as reviewed was inappropriate, and we will provide the figure with a proper resolution once we can upload the file.

-Line 1: Should be “genomes…”, not “genome…”.

- We have modified the text accordingly.

-Line 133: Is it GC skew or GC-skew?

- We have modified the text accordingly, using GC-skew in all mentions.

-Line 159: Is the phylogenetic tree constructed using 13 amino acid sequences or 13 PCGs? According to line 177, the tree was constructed using the PCGs. 

- The phylogenetic analyses were performed with the amino acid sequences of the 13 PCGs. We have modified the text to clarify the points mentioning the matrix used in the phylogenetic reconstructions.

-Lines 178-179: Replace “.” with “,”.

- We have modified the text accordingly.

REVIEWER #3

1. The results of the phylogenetic analysis should be shown in "Abstract".

- As requested, we have included the part of the phylogenetic analysis in the abstract.

2. The table header is missed in "Table 3".

- We have corrected Table 3 putting the headings that were missing.

3. In the list of references, the format of the journal names is inconsistent. Please check and make revisions as required by the journal format.

- We have checked and corrected the references, according to the journal format.

4. The English language of this manuscript needs to be revised for clarity. For example:

(1) Line 28, Suggested reword "and each comprises......"

(2) Line 96, Suggested reword "......Finally, we assessed the"

(3) Line 98, Suggested reword "on the amino acid supermatrix of the 13 PCGs."

(4) Line 127, Suggested reword "checked and curated in Geneious. The tRNAs ......"

(5) Line 171, Suggested reword "...... the BI tree was"

(6) Line 178, Suggested reword "...... of Furipterus horrens, "

(7) Line 179, Suggested reword "Lonchorhina aurita, Natalus macrourus, and 54 other bat genera with one species each, "

(8) Line 181, Suggested reword "......black PP = 1.00; red "

(9) Line 182, Suggested reword "......and white 0.95 > PP ≥ 0.90. The triangles ...... "

(10) Line 183, Suggested reword "...... approach: black"

(11) Line 324, Suggested reword "The Chiroptera was grouped ......"

- We have carefully revised the manuscript and corrected all issues related to the English language, including all points mentioned by the reviewer. 

In addition to the corrections indicated by the reviewers, we also included other modifications, which are also highlighted in the text, aiming to improve the manuscript.

Thank you very much for your kind attention.

Sincerely yours,

Santelmo Vasconcelos

Instituto Tecnológico Vale

Rua Boaventura da Silva 955

66055-090 Belém, Pará, Brazil

Phone: +55 91 3213 5400

santelmo.vasconcelos@itv.org

---

## [Decision Letter · Decision Letter 2]

30 Jul 2024

Complete mitochondrial genomes of three vulnerable cave bat species and their phylogenetic relationships within the order Chiroptera

PONE-D-24-05101R2

Dear Dr. Vasconcelos,

We’re pleased to inform you that your manuscript has been judged scientifically suitable for publication and will be formally accepted for publication once it meets all outstanding technical requirements.

Kind regards,

Pankaj Bhardwaj, Ph.D.

Academic Editor

PLOS ONE

Additional Editor Comments (optional):

Reviewers' comments:

Reviewer's Responses to Questions

**Comments to the Author**

1. If the authors have adequately addressed your comments raised in a previous round of review and you feel that this manuscript is now acceptable for publication, you may indicate that here to bypass the “Comments to the Author” section, enter your conflict of interest statement in the “Confidential to Editor” section, and submit your "Accept" recommendation.

Reviewer #1: All comments have been addressed

Reviewer #3: All comments have been addressed

2. Is the manuscript technically sound, and do the data support the conclusions?

Reviewer #1: Yes

Reviewer #3: Yes

3. Has the statistical analysis been performed appropriately and rigorously? 

Reviewer #1: Yes

Reviewer #3: Yes

4. Have the authors made all data underlying the findings in their manuscript fully available?

Reviewer #1: Yes

Reviewer #3: Yes

5. Is the manuscript presented in an intelligible fashion and written in standard English?

Reviewer #1: Yes

Reviewer #3: Yes

6. Review Comments to the Author

Reviewer #1: Dear Editor,

The present version of the revised manuscript is well written and addressed all comments and suggestions. I would like to accept the manuscript in the present form and have the following two suggestions before formal publications:

Page 55 - Line 232-233

Kindly correct this phrase“with 29 genes (12 PCGs, the two rRNAs and 14 tRNAs) being located in the heavy (H)”. Here only 28 genes are present.

Page 62 - Line 357

Correct this format “S3-S5 Tables” as Tables S3-S5

Regards,

Reviewer #3: 1. Line 206, replace "29" with "28"

2. Line 209, The overall nucleotide composition of mitogenome is not included in Table 3.

3. Line 227,348 and 352, replace "mitochondrial genomes" with "mitogenomes"

4. Line 260, replace "NAD4" with "nad4"

5. Line 291, The gene names in figure 2 should be consistent with the text of this manuscript.

6. Line 410 and 510, replace "." with ","

7. PLOS authors have the option to publish the peer review history of their article (what does this mean?). If published, this will include your full peer review and any attached files.

Reviewer #1: **Yes: **Muhammad Asghar Hassan (Institute of Entomology, Guizhou University, Guiyang, China)

Reviewer #3: No

---

## [Editor Report · Acceptance letter]

13 Aug 2024

PONE-D-24-05101R2 

PLOS ONE

Dear Dr. Vasconcelos, 

I'm pleased to inform you that your manuscript has been deemed suitable for publication in PLOS ONE. Congratulations! Your manuscript is now being handed over to our production team.

Kind regards, 

on behalf of

Dr. Pankaj Bhardwaj 

Academic Editor

PLOS ONE